
**Hydrological Drought forecasting under changing environment**
**in Luanhe River basin**
Min Li[1,2], Mingfeng Zhang [2], Runxiang Cao [3], Yidi Sun[2], Xiyuan Deng [4,5],
[1] State Key Laboratory of Hydraulic Engineering Simulation and Safety , Tianjin
University, Tianjin, China;
[2] College of Hydraulic Science and Engineering, Yangzhou University, JiangSu,
China;
[3] College of Water Resources, North China University of Water Resources and Electric
Power, Zhengzhou 450046, China;
[4] Nanjing Hydraulic Research Institute, Nanjing, 210029, China; [5] State Key
Laboratory of Hydrology-Water Resources and Hydraulic Engineering , Nanjing,
210029, China
Correspondence to: Min Li ( limintju@126.com); Xiyuan Deng (xydeng@nhri.cn)



**Abstract:** Hydrological drought forecasting can mitigate the socio-economic and ecological impacts of drought. It is an important disaster reduction strategy to forecast the occurrence of hydrological drought according to the forecasting system. In this paper, the conditional distribution model with human activity factor as exogenous variable was constructed to forecast the hydrological drought based on meteorological drought, and then compared with the traditional normal distribution model and conditional distribution model. The results show that the runoff series of Luanhe River Basin from 1961 to 2010 was non-stationary. For the traditional conditional probability models, the transition probabilities of drought were affected by SPI time scales and forecasting periods. In order to analyze the impact of human activities on hydrological drought, we constructed the human activity factor based on the method of restoration. Subsequently, the conditional distribution models involving human index were constructed and the influence of human activities on drought transition probability was analyzed. With the increase of human index ($HI$) value, hydrological droughts tend to transition to more severe droughts. Finally, a scoring mechanism was applied to evaluate the performance of three drought forecasting models. According to the scores of the three drought forecasting models, the conditional distribution model involving of human activity factor can further improve the forecasting accuracy of drought in Luanhe River Basin.

**Keyword:** Changing environment; Drought forecasting; Human activity factor; Luanhe River basin

## 1 Introduction

Typically, meteorological drought is regarded as the beginning of a drought event; after the occurrence of meteorological drought, other drought phenomena occur, such as hydrological drought (Miriam et al., 2018; Fuentes et al., 2022; Wang et al., 2021). However, there is a delay period from meteorological drought to hydrological drought (Ding et al., 2021; Xu et al., 2019). Therefore, the occurrence of hydrological drought can be forecasted according to meteorological drought monitoring. Accurate hydrological forecast information is beneficial to reduce the losses caused by hydrological drought. (Behzad and Hamid, 2019; Melanie et al., 2018).

Statistical technology is an effective prediction method that has been widely used in drought forecasting in recent years (Bonaccorso et al., 2015). Focusing on statistical techniques, several mathematical statistical models have been applied to forecast drought, such as neural network models (Mehdi et al., 2016; Maryam et al., 2017; Ahnadi et al., 2011), time series modelling (Mohammad et al., 2020; Natsagdorj





et al., 2021; Stojković et al., 2020) and hybrid models (Alquraish et al., 2021; Abbasi
et al., 2021; Bagher et al., 2013). Some scholars focus on the transition probability of
the drought class, which is mainly based on a certain drought index, such as the
standardized precipitation index (SPI), the Palmer drought severity index (PDSI) or
the standardized runoff index (SRI) (McKee, 1993; Palmer, 1965; Shukla, 2008).
Mallya et al. (2013) assessed a drought probability based on a hidden Markov model
(HMM) and then analysed the drought characteristics of Indiana. Moreira et al. (2013)
calculated the SPI time series in the Alentejo area from 1932 to 1999, and then
loglinear models were fitted to assess drought class transition probabilities. Based on
the Multivariate Standardized Precipitation Index (MSPI), Aghelpour and Varshavian
(2021) proposed the hybrid model to forecast the hydrological drought in Iran, which
significantly improved the forecasting accuracy. Majid et al. (2019) used
Archimedean copulas to model the relationship between the SPI and standardized
hydrological drought index (SHDI), and the results indicated that hydrological
drought class forecasting in the coming month is promising with less than 10% error.
Considering the impact of the changing environment, Bonaccorso et al. (2015)
calculated SPI values under distinct time scales and analysed the conditional
probabilities from the current SPI values to the future SPI classes. Ren et al. (2017)
found that a model using large-scale climatic indices as covariates can improve the
accuracy of meteorological drought forecasting in the Luanhe River Basin. Although
some progress has been made in the study of drought forecasting, there are few
studies considering the impact of changing environments.
To date, some studies have found nonstationary characteristics in the
hydrological series of the Luanhe River Basin under changes in the environment
(Wang et al., 2018; Li et al., 2015; Wang et al., 2016). The nonstationarity of
hydrological series may lead to the nonstationarity of the relationship between
hydrological series (for example, precipitation and runoff series), and traditional
drought prediction methods are no longer applicable (Wang et al.,2022; Dixit et
al.,2022; Muhammad et al.,2020; Zhao et al.,2018; Charles, 2017; Carmelo and Jü




rgen, 2018).
The research contents of this paper are as follows: (1) The SPI series and SRI
series are calculated according to the monthly rainfall and runoff data of the Luanhe
River Basin from 1961 to 2010. (2) A multivariate normal distribution model (Model
1), conditional distribution model (Model 2) and conditional distribution model with
the human index (*HI*) as an exogenous variable (Model 3) were constructed to
calculate the transition probabilities from current SPI classes or values to future SRI
classes. (3) A scoring mechanism was applied to the evaluation of the three
probability models.
In addition to the introduction, this paper also contains the following sections.
Section 2 introduces the study area and data. Section 3 briefly describes the methods
used in the research. Section 4 introduces the model construction and calculation
results and analyses the results. Section 5 presents the prospects.
**2 Study area and data**
The Luanhe River Basin, located in the subtropical monsoon region, covers an
area of approximately 33700 square kilometres. Its geographical location is shown in
Figure 1. Due to the influence of geographical location and topography, the annual
average north-south temperature difference in the basin is 11.5 °C, and the annual
rainfall distribution is uneven. Less rain in spring and winter makes the area prone to
meteorological drought and hydrological drought, while there is relatively more
rainfall in summer. The average rainfall in summer is approximately 200-560 mm,
resulting in highly variable annual runoff of the basin. The concentrated rainfall in
summer has also become one of the remarkable features of the climate in this area. In
recent years, the precipitation and inflow of the Luanhe River Basin have gradually
decreased, the water level of the Panjiakou Reservoir in the lower reaches of the basin
has decreased, the runoff has also decreased, and the frequency of meteorological
drought and hydrological drought has significantly increased. Especially after entering
the 21st century, the river basin has exhibited the phenomenon of continuous drought
and even extreme drought. With the change in the global climate and the impact of



human activities on the basin environment, drought disasters in the Luanhe River
Basin occur frequently, causing significant social and economic losses.

In this paper, the monthly rainfall data from 26 stations in the Luanhe River

Basin from 1961 to 2010 are provided by the Hebei Provincial Hydrology and Water
Resources Investigation Bureau. The average monthly rainfall data of the area are
obtained by spatial interpolation. The runoff data from 1961 to 2010 come from the
inflow runoff series of the Panjiakou Reservoir. The SPI and SRI can be calculated for
1-month, 3-month, 6-month, and 12-month time scales to characterize meteorological
drought and hydrological drought based on these data.

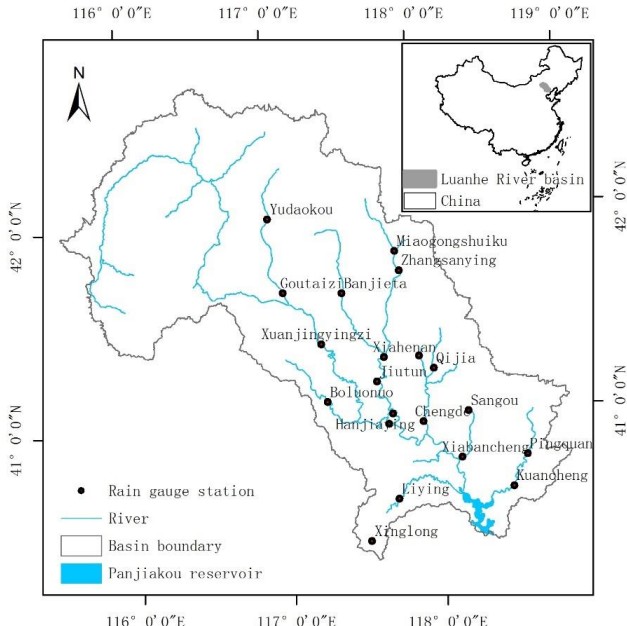


Figure 1 The geographical location of the Luanhe River Basin

**3 Methods**
3.1 Nonstationarity test method

In the case of environmental changes, nonstationarity may occur in hydrological

series. The Pettitt test, as one of the important methods to test whether there is
nonstationarity in time series, can identify whether there are change points in the


sample series (Malede et al., 2022). Assuming that the sample sequence is
$x = (x_1, x_2, \cdots x_n)$, the formula is as follows:
$$U_{t,n} = U_{t-1,n} + \sum_{i=1}^{n} \mathrm{sgn}(x_t - x_i) \quad (t = 2, 3, \cdots n)t_0 \qquad (1)$$

$$\mathrm{sgn}(x_t - x_i) = \begin{cases} 1 & x_t - x_i > 0 \\ 0 & x_t - x_i = 0 \\ -1 & x_t - x_i = 0 \end{cases} \qquad (2)$$

where $U_{t,n}$ is the test statistic, which indicates the cumulative number of the
values at time t greater than or less than the values at time i. In addition, if $K_{t0,n}$
satisfies:
$$K_{t0,n} = \max |U_{t,n}| \quad (t=1,2,\cdots,n) \qquad (3)$$

Then, $t_0$ is considered to be the change point, and the cumulative probability of
possible change is determined by $K_{t0,n}$:
$$P_{t0,n} = 2\exp(-\frac{6K_{t0,n}^2}{n^3 + n^2}) \qquad (4)$$

Given the significance level $\alpha = 0.05$, if $P_{t0,n} > 0.95$, it means that the point is a
significant change point (Li et al., 2022; Koudahe et al., 2018). Furthermore,
combined with the Mann-Kendall test, the trend characteristics of the sample series
can be obtained (Linchao et al., 2018).
The sliding T test is a basic method commonly used in statistics. According to
the mean and variance of the two sample sequences before and after the change points
in the runoff time series, the two sample sequences are tested (Li et al., 2020):
$$t = \frac{\overline{x}_1 - \overline{x}_2}{S\sqrt{\frac{1}{n_1} + \frac{1}{n_2}}} \qquad (5)$$

$$S = \sqrt{\frac{(n_1 - 1)S_1^2 + (n_2 - 1)S_2^2}{n_1 + n_2 - 2}} \qquad (6)$$

$$S_1^2 = \frac{1}{n_1 - 1}\sum_{t=1}^{n_1}(x_t - \overline{x}_1)^2 \qquad (7)$$

$$S_2^2 = \frac{1}{n_2 - 1}\sum_{t=1}^{n_1+n_2}(x_t - \overline{x}_2)^2 \qquad (8)$$





Here, assume that the change point is $x_t$, $n_1$ and $n_2$ represent the sample size
before and after the change point, $S_1^2$ and $S_2^2$ represent the variance of the samples
before and after the change point, respectively If the statistic $t$ satisfies $t > t_\alpha$ as the
significance level is $\alpha = 0.05$, the point can be considered the change point.
The Spearman correlation test can be applied to test the trend of time series, and
the specific description refers to the article of Bishara and Hittner (2012).
3.2 Human activity index (*HI*)
The double cumulative curve method can test the nonstationarity of the bivariate
correlation between rainfall series and runoff series, and the point where the
underlying surface is significantly altered by human activities can be determined
according to the position of the slope change of the curve. The linear regression
relationship of the cumulative rainfall and runoff series can be calculated according to
the following formula:
$$\sum x = k \sum y + b \qquad (9)$$

Here, $x$ is the runoff series; $y$ is the rainfall series; $k$ is the correlation coefficient of the
regression equation; and $b$ is the intercept of the regression equation.
Human activities are the main reason for the nonstationarity of the runoff series
in the watershed, so the human activity index (*HI*) can be constructed to quantify the
impact of human activities on runoff. Based on the linear regression relationship
established between the accumulated precipitation and the accumulated runoff before
the change point, the theoretical runoff sequence during the human activity period can
be calculated from the measured precipitation sequence. $\mathrm{SRI}'$ represents the
standardized runoff index value without human activity interference, and $\mathrm{SRI}$
represents the normalized runoff index value calculated based on the measured runoff
sequence under the disturbance of human activities. The *HI* is obtained by subtracting
the theoretical $\mathrm{SRI}'$ and the actual $\mathrm{SRI}$, and the calculation formula is as follows:
$$HI = SRI' - SRI \qquad (10)$$



When *HI*>0, it can be assumed that human activities exacerbate hydrological
drought, *HI*<0 has the opposite effect, and *HI*=0, the watershed is considered
undisturbed by human activities.
3.3 Multivariate normal distribution model
The SPI is one of the important indicators for evaluating meteorological drought
in the basin, and the SRI is an important indicator for evaluating hydrological drought
in the basin. According to the rainfall data and runoff data in the basin, the SPI and
SRI at different time scales can be calculated. Table 1 provides the drought class
classification and corresponding SPI values and SRI values (Kolachian and Saghafian,

2021).

Table 1 Drought class classification and corresponding SPI values and SRI values

| SPI/SRI values | Class |
|---|---|
| > -0.99 | Normal |
| -1.00 to -1.49 | Moderate |
| -1.50 to -1.99 | Severe |
| ≤ -2.00 | Extreme |


As a traditional drought class forecasting model, the multivariate normal
distribution model (Model 1) can forecast the future SRI class according to the current
SPI class. Assuming that both the current SPI and SRI sequence satisfy a
multivariable normal distribution, the joint probability density can be expressed as
follows (Chang et al.,2022):
$$f_{Z_{v,\lambda}^{(k)}W_{v,\lambda+M}^{(k)}}(t,s) = \frac{1}{2\pi|\Sigma|}\cdot\exp\left(-\frac{1}{2}X^T\Sigma^{-1}X\right) \qquad (11)$$

Here, $\sum$ is the covariance matrix, and $X = \begin{bmatrix} t, s \end{bmatrix}^T$. The form of the covariance
matrix is as follows:
$$\Sigma = \begin{bmatrix} 1 & \text{cov}\left[Z_{v,\lambda}^{(k)},W_{v,\lambda+M}^{(k)}\right] \\ \text{cov}\left[Z_{v,\lambda}^{(k)},W_{v,\lambda+M}^{(k)}\right] & 1 \end{bmatrix} \qquad (12)$$

Furthermore, according to the joint probability density function of the SPI value
$Z_{v,\lambda}^{(k)}$ at $v$ year and month $\lambda$ and the future $M$ months SRI value $W_{v,\lambda+M}^{(k)}$, the analytical





formula of the transition probability of the future SRI drought class can be obtained
(Zhang et al., 2017):
$$P\left[W_{v,\lambda+M}^{(k)} \in C_M\right] = \frac{\iint_{C_N C_M} f_{Z_{v,\lambda}^{(k)} W_{v,\lambda+M}^{(k)}}(t,s) \cdot dt \cdot ds}{\int_{C_N} f_{Z_{v,\lambda}^{(k)}}(t) \cdot dt}$$
(13)

where $C_M$ represents the drought class and $f_{Z_{v,\lambda}^{(k)}}(t)$ represents the marginal
density function of $Z_{v,\lambda}^{(k)}$ in the current $\lambda$ month.
3.4 The conditional distribution model
The conditional distribution model (Model 2) proposed by Bonaccorso et al.
(2015) is described as follows: when one group of sample data $X$ obeys a normal
distribution and satisfies $X \sim N(\mu_1, \Sigma_1)$, while another group of sample data $Y$ also
obeys a normal distribution, namely, $Y \sim N(\mu_2, \Sigma_2)$, then the total sequence can be
written as follows:
$$B = \begin{bmatrix} X \\ Y \end{bmatrix} \begin{matrix} r \\ p-r \end{matrix} \sim N_p\left(\begin{bmatrix} \mu_1 \\ \mu_2 \end{bmatrix}, \begin{bmatrix} \Sigma_{11} & \Sigma_{12} \\ \Sigma_{21} & \Sigma_{22} \end{bmatrix}\right)$$
(14)

When sequence $Y$ obeys a normal distribution, the distribution of sequence $X$
under the $Y$ condition still satisfies a normal distribution, namely, the distribution of
$(X \mid Y)$ is as follows (Gong et al. 2021):
$$(X \mid Y) \sim N(\mu_3, \Sigma_3)$$
(15)

where $\mu_3$ represents the expected value under the conditional distribution, and
$\Sigma_3$ is the conditional covariance matrix:
$$\mu_3 = \mu_1 + \Sigma_{12}\Sigma_{22}^{-1}(y - \mu_2)$$
(16)

$$\Sigma_3 = \Sigma_{11} - \Sigma_{12}\Sigma_{22}^{-1}\Sigma_{21}$$
(17)

Assuming that the current SPI and the SRI sequence that transitioned from
meteorological drought satisfy a binary normal distribution, then the probability of the
transition to the future SRI drought class under the current SPI value can be deduced
as follows (Ren et al., (2017)):





$$P\left[W_{v,\lambda+M} \in C_M / Z_{v,\lambda} = z_0\right] = \int_{C_{Mi}}^{C_{Ms}} \frac{1}{\sqrt{2\pi}\sigma_Z} e^{-\frac{1}{2}\left(\frac{x-\rho z_0}{1-\rho^2}\right)^2} dx \qquad (18)$$

where $Z_{v,\lambda}$ represents the SPI value of the current month $\lambda$, $W_{v,\lambda+M}$ represents the
SRI value of the $\lambda + M$ month, $C_{Ms}$ and $C_{Mi}$ are the upper and lower limits of the
drought class $C_M$, and the correlation coefficient between the current SPI value and
the future SRI value is $\rho$. Furthermore, the current SPI and future SRI can be
expressed as the standard normal cumulative distribution function $\Phi$:
$$P\left[W_{v,\lambda+M} \in C_M \mid Z_{v,\lambda} = z_0\right] = \Phi\left[\frac{C_{Ms} - \rho \bullet z_0}{1-\rho^2}\right] - \Phi\left[\frac{C_{Mi} - \rho \bullet z_0}{1-\rho^2}\right] \qquad (19)$$

The calculation of the correlation coefficient $\rho$ is as follows:
$$\rho = \frac{\mathrm{cov}[Z_{v,\lambda}^{(k)}, W_{v,\lambda+M}^{(k)}]}{\sqrt{\mathrm{var}(Z_{v,\lambda}^{(k)})\,\mathrm{var}(W_{v,\lambda+M}^{(k)})}} \qquad (20)$$

$K$ represents the time scale of the drought index. Assuming that the cumulative
rainfall $Y$ and runoff $X$ satisfy a normal distribution, then after the standardization
process, the SPI value $Z_{v,\lambda}^{(k)}$ corresponding to cumulative rainfall $Y$ and SRI value
$W_{v,\lambda+M}$ corresponding to runoff $X$ obey the standard normal distribution (Wu, 2019),
namely:
$$\mathrm{var}(Z_{v,\lambda}^{(k)}) = \mathrm{var}(W_{v,\lambda+M}^{(k)}) = 1 \qquad (21)$$

$\mathrm{cov}[Z_{v,\lambda}^{(k)}, W_{v,\lambda+M}^{(k)}]$ represents the covariance between the current SPI and the Sri
value with a forecast period of M months. The calculation is as follows:
$$\mathrm{cov}[Z_{v,\lambda}^{(k)}, W_{v,\lambda+M}^{(k)}] = \frac{1}{\sqrt{\sum_{i=0}^{k-1}\sigma_{\lambda+M-i}^2 \sum_{j=0}^{k-1}\sigma_{\lambda-j}^2}} \cdot \sum_{i=0}^{k-1}\sum_{j=0}^{k-1}\mathrm{cov}\left[X_{v,\lambda+M-j}, Y_{v,\lambda-i}\right] \qquad (22)$$

3.5 The conditional distribution model involving *HI* as an exogenous variable
According to the above conditional probability model, when considering *HI* as
an exogenous variable, the model (Model 3) can be extended as follows:



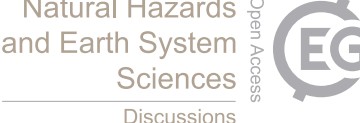
$$P\left[W_{v,\lambda+M} \in C_M / Z_{v,\lambda} = z_0, H_{v,\lambda} = h_0\right] = \int_{C_{Mi}}^{C_{Ms}} \frac{1}{\sqrt{2\pi}\sigma_z} e^{-\frac{1}{2}\left(\frac{x-\mu_z}{\sigma_z}\right)^2} dx \qquad (23)$$

$$\mu_z = E\left[W_{v,\lambda+M} \mid Z_{v,\lambda}, H_{v,\lambda}\right] = \Sigma'_{12}(\Sigma'_{22})^{-1}\begin{bmatrix} z_0 \\ h_0 \end{bmatrix} \qquad (24)$$

$$\sigma_z^2 = \mathrm{var}\left[W_{v,\lambda+M} \mid Z_{v,\lambda}, H_{v,\lambda}\right] = 1 - \Sigma'_{12}(\Sigma'_{22})^{-1}\Sigma'_{21} \qquad (25)$$

Where:

$$\Sigma'_{12} = \left[\mathrm{cov}(W_{v,\lambda+M}, Z_{v,\lambda}) \ \ \mathrm{cov}(W_{v,\lambda+M}, H_{v,\lambda})\right] \qquad (26)$$

$$\Sigma'_{22} = \begin{bmatrix} 1 & \mathrm{cov}(Z_{v,\lambda}, H_{v,\lambda}) \\ \mathrm{cov}(H_{v,\lambda}, Z_{v,\lambda}) & 1 \end{bmatrix} \qquad (27)$$

$$\Sigma'_{21} = (\Sigma'_{12})^T \qquad (28)$$

## 4 Results and discussion

4.1 Nonstationarity analysis

In this paper, the area average monthly rainfall data of the Luanhe River Basin from 1961 to 2010 are obtained by spatial interpolation. The runoff data come from the inflow runoff series of the Panjiakou Reservoir. Given the significance level $\alpha = 0.05$, the nonstationarity test results are shown in Figure (2).

Figure 2 (a) shows that the years of possible runoff change were 1979, 1996, 1997, 1998, and 1999. The P values in 1979 and 1998 were infinitely close to 1, which were considered to be extremely significant runoff change points. Among all the possible points satisfying $t > t_\alpha$, there are two maximum points (Figure 2 (b)), namely, 1979 and 1998, which are considered to be possible runoff change points. The final change point needs to be judged based on the actual situation of the watershed.

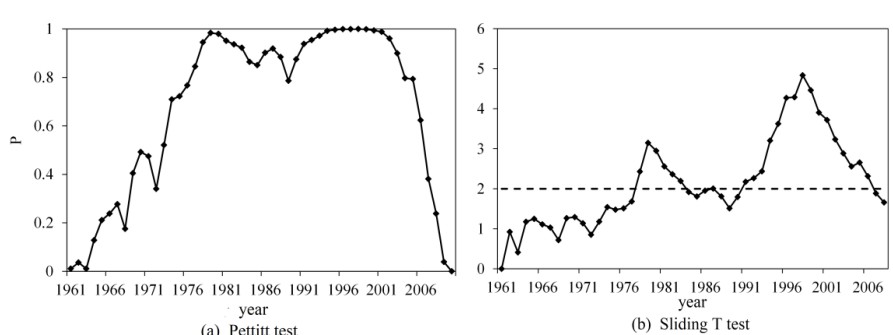

(a) Pettitt test        (b) Sliding T test

Figure (2)    The change points of the runoff series

The results of the Spearman correlation test (Table 2) indicate that the runoff series showed an upwards trend before 1979, but the trend was not significant. However, there was a significant downwards trend in the series after 1979. In general, the runoff series showed a significant downwards trend.

Table 2. Spearman correlation test results of runoff series trend

| Runoff series | statistic $t$ | Critical value $t_\alpha$ |
|---|---|---|
| The whole series | -3.471 | ±2.009 |
| Serie before 1979 | 0.691 | ±2.009 |
| Serie after 1979 | -2.292 | ±2.009 |

In addition, according to historical records, there were no extreme rainstorm events recorded during 1979. It can be inferred that the cause of the sudden change in annual runoff in 1979 was not the formation of heavy rainstorms in the previous period or the same period. Since the start of 1979, the underlying surface conditions of the basin have undergone large changes due to human activities, so it is determined that 1979 is the change point of the runoff sequence in the basin. Therefore, 1979 was finally determined as the change point of the runoff sequence of the Luanhe basin from 1961 to 2010. This conclusion is consistent with Li et al. (2015) and Wang et al. (2015).

4.2 Transition probabilities from current SPI values to future SRI classes

According to the normality test results of rainfall and runoff series, it is reasonable to apply the conditional distribution model. To analyse the influence of different time scales of SPI on the transition probabilities, using the forecast period as



one month and the time scales of SPI on 1-month, 3-month, 6-month and 12-month as
examples, the probabilities of converting SPI values to SRI classes were calculated
(Figure (3)).

As shown in Figure (3), when meteorological drought is categorised as extreme

drought, the probability of maintaining the SRI class in the extreme drought state is
greater as the time scale of the SPI increases. As the SPI is a 12-month time scale, the
drought transition probability is close to 1. However, while the time scale is small, the
response of the future SRI value to rainfall is faster, so the probability of tending to
the normal state is greater. In the future, the response of the SRI value to rainfall is
relatively fast, so it is more likely to tend to a normal state.

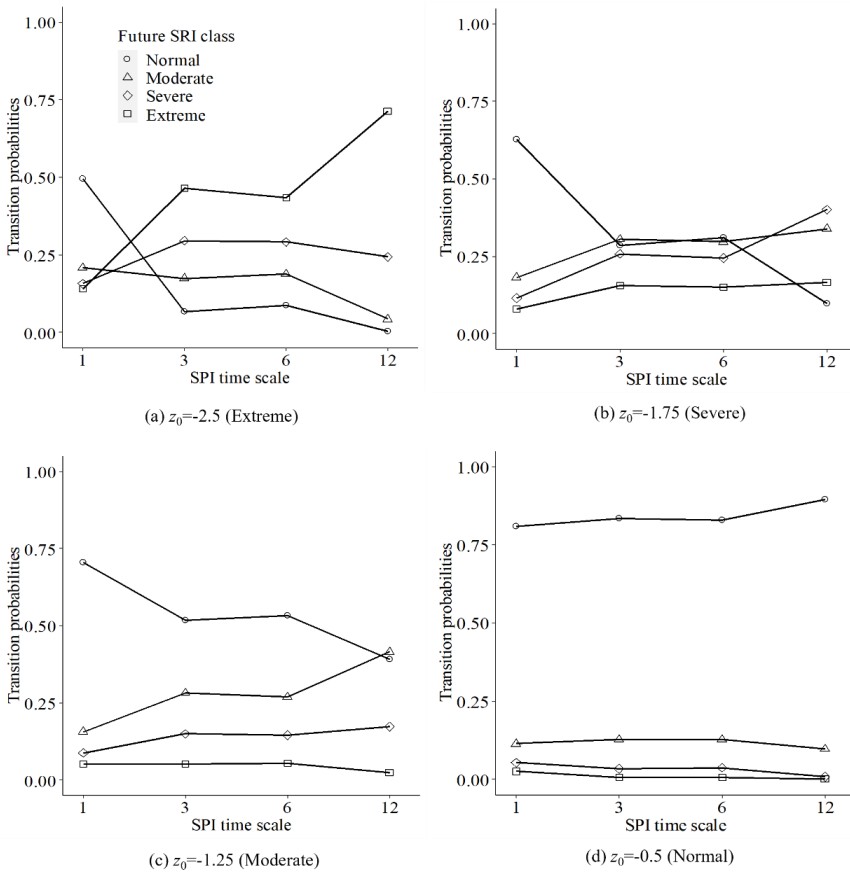


Figure (3). Influence of the SPI time scale on transition probabilities ($z_0$ : initial value of SPI)





In addition, the transition probabilities of drought are distinct for different
forecast periods. As seen in Figure 4(a), while the current period $z_0 = -2.5$, i.e., the
meteorological drought is extreme drought and the forecast period is 1 and 2 months,
the probability of its future SRI class being extreme drought is the highest. Moreover,
the probability of its future SRI drought class returning to normal status becomes
higher as the forecast period becomes longer. When the current period $z_0 = -1.75$
(Figure 4 (b)), namely, the meteorological drought is severe drought and the forecast
period is 1 month, its future SRI class tends to be normal or moderate drought. While
the forecast period becomes longer, its drought degree gradually decreases and tends
to be normal. When the current period $z_0 = -1.25$ (Figure 4 (c)), namely, the
meteorological drought is a moderate drought, the future SRI class tends to be a
normal or moderate drought state as the forecast period is 1 month. In addition, its
drought degree gradually decreases and tends to be normal, while the forecast period
becomes longer. It is worth noting that the current $z_0 = 0$ (Figure 4 (d)), and the
probability that the future SRI class is normal as the forecast period is 1, 2 and 3
months is greater than 0.8.



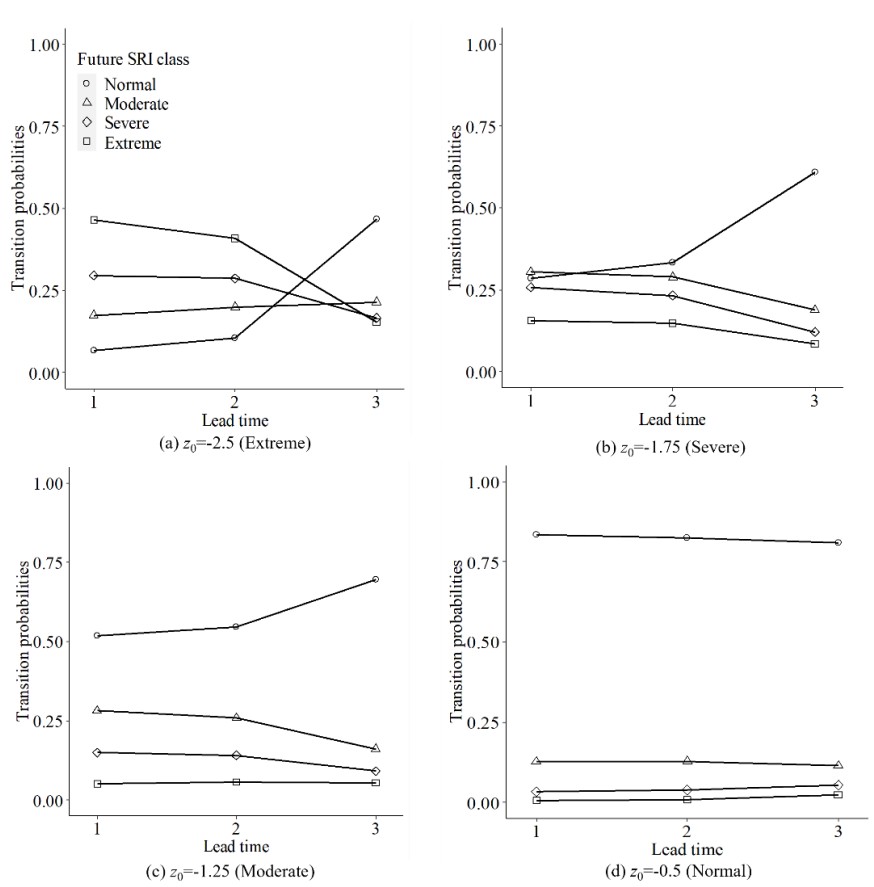

(a) $z_0$=-2.5 (Extreme)

(b) $z_0$=-1.75 (Severe)

(c) $z_0$=-1.25 (Moderate)

(d) $z_0$=-0.5 (Normal)

Figure (4) Influence of forecast period on transition probabilities ( $z_0$ :initial value of SPI)

From the above analysis, when the forecast period is short (*M*=1 or 2), the hydrological drought class obtained from the transition of meteorological drought tends to be the same as that of meteorological drought. With the extension of the forecast period (*M*=2 or 3), the overall SRI class obtained from the transition tends to be slightly lower than the SPI drought class or to the normal state, i.e., the hydrological drought class obtained from the transition tends to be slightly lighter than the meteorological drought on the whole or to be maintained in the normal state.

4.3 Transition probabilities with involving *HI* as the covariate

According to the above nonstationarity test results, 1979 was the change point, and the linear regression relationship of the cumulative rainfall and runoff series





before and after the change point were established. The calculation results are shown
in Table 3:
Table 3 Linear regression relationship between cumulative precipitation ($x/\mathrm{mm}$) and cumulative
runoff ($y/10^6\mathrm{m}^3$)

| Period | Linear regression equation | Correlation coefficient |
|---|---|---|
| 1961~1979 | $x = 0.0276\,y + 2.7566$ | 0.99 |
| 1980~2010 | $x = 0.0307\,y - 30.652$ | 0.98 |

The HI results for different time scales are shown in Figure 5.

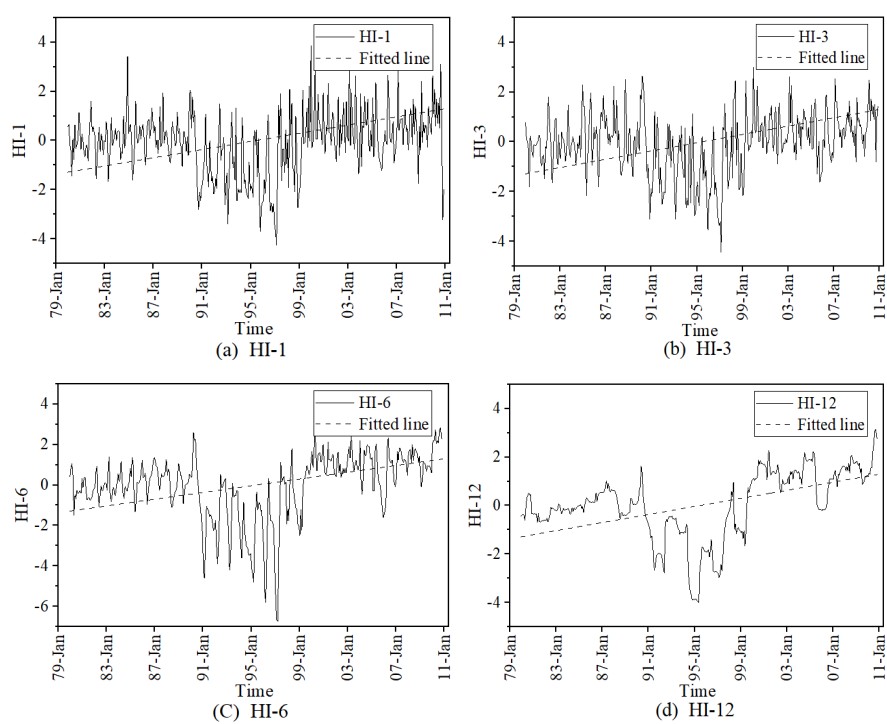

(a) HI-1    (b) HI-3    (C) HI-6    (d) HI-12


Figure 5 Different average periods of *HI* (*HI*-1: *HI* with 1-month time scale; *HI*-3: *HI* with 3-
month time scale; *HI*-6: *HI* with 6-month time scale; *HI*-12: *HI* with 12-month time scale)
As shown in Figure 5, the *HI* at all monthly scales generally ranges upwards,
which means that human activities have intensified the occurrence of hydrological
drought.
The *HI* of different monthly scales were standardized, taking the 12-month time
scale as an example, and the results were calculated as shown in Table 3.
Table 3. *HI*-12 Monthly Mean and Standard Deviation


|  | J | F | M | A | M | J | J | A | S | O | N | D |
|---|---|---|---|---|---|---|---|---|---|---|---|---|
| Mean | -0.04 | -0.03 | -0.03 | -0.03 | -0.03 | 0.00 | 0.06 | 0.06 | 0.10 | 0.10 | 0.09 | 0.06 |
| Sd | 1.36 | 1.37 | 1.38 | 1.41 | 1.41 | 1.51 | 1.40 | 1.40 | 1.45 | 1.44 | 1.44 | 1.43 |

Furthermore, the drought transition probabilities involving *HI* can be calculated
from Eq. (23). Using the forecast period of one month from December and the SPI
time scale of 12 months as an example, the drought transition probabilities from
current SPI values to future SRI classes can be calculated (Figure 6). To analyse the
effect of human activities on the drought transition probability more clearly, the
calculation results of the three models are compared here separately. The horizontal
coordinate indicates the drought classes corresponding to the SRI for the coming
month, and the vertical coordinate is the drought transition probability.

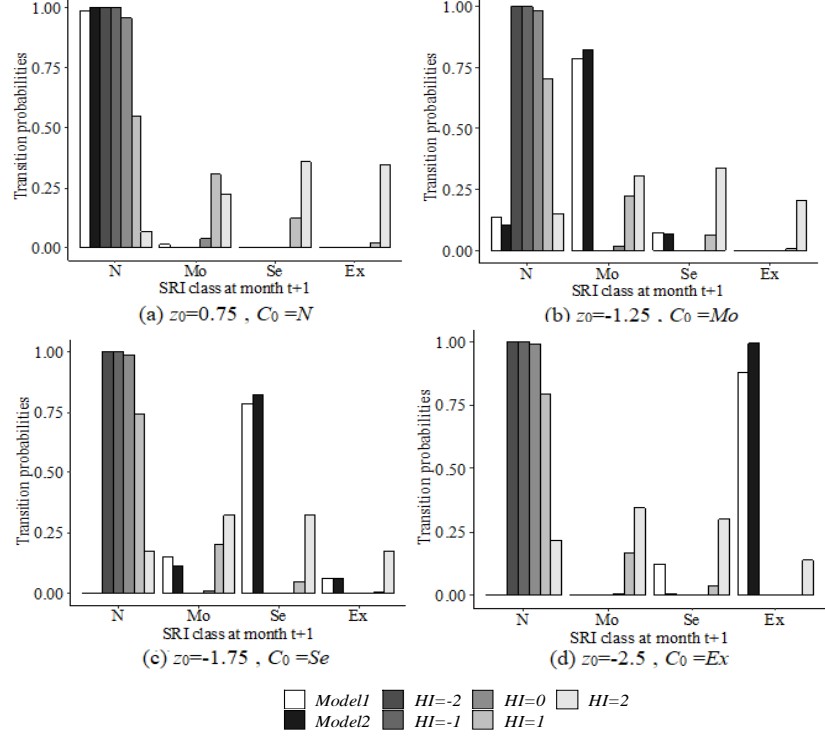

Figure 6 Drought transition probability under the influence of human activities ($C_0$ denotes the
initial drought class of SPI in the multivariate normal model; $z_0$ represents the initial value of SPI
in the conditional distribution model; Model 1: The normal distribution model; Model 2: The
conditional distribution model involving *HI*)





In Figure 6 (a), when the initial $z_0$=0.75 and $c_0$=N, the results shown in Model 1
and Model 2 are similar, and the probability transitions of SPI values to SRI classes in
the future month in the normal class are close to 1. However, the results of Model 3
indicate that the probabilities of maintaining SRI in the normal class in the future
decrease as *HI* increases. When *HI*=2, the probability of transition to severe drought
or extreme drought is higher.
From the initial $z_0$=-1.25 and $c_0$=Mo (Figure 6 (b)), it can be seen from the
results of Model 3 that the transition probabilities of SPI values to a normal SRI class
in the coming month are higher when *HI* is less than 1. As the *HI* increases, the
transition probabilities of the SPI values to a moderate drought or even a more severe
drought in the future increase. In addition, the probabilities of maintaining drought at
moderate drought are the highest when human activities are not considered, and
Model 2 shows a higher probability.
While the initial meteorological drought class is a severe drought (Figure 6 (c)),
the probabilities of the future SRI drought class being in the normal class become
larger as the *HI* decreases. When the effect of human activities is not considered, the
probability that the current SPI value transitions to the SRI class under severe drought
in the future month is the highest, and the probability of being in the normal class is
the lowest. For Model 1, the probability of the SRI classes transitioning to a moderate
drought is higher than the result of Model 2.
It is noteworthy that when the initial $z_0$=-2.5 and $c_0$=Es (Figure 6 (d)), the
probabilities of transition of the SPI values to future SRI classes at the normal class
are close to 1 as *HI*<0. However, hydrological drought is more likely to be moderate
drought or severe drought as *HI* are greater than 0, and the transition probabilities
exceed 0.25. For Model 1 and Model 2, the probabilities of transition of current SPI
values or classes to the future month SRI classes also in extreme drought are both
higher than 0.75. Model 1 shows a higher probability than Model 2 when the SRI
class transitions to severe drought.



4.4 Model evaluation and analysis

To quantitatively evaluate the prediction accuracy of Model 1, Model 2 and

Model 3, the study period is divided into a correction period (1961-2003) and a
verification period (2004-2010), and then the drought transition probability from the
SPI value or class to the SRI class in the future M-month is calculated. The monthly
drought transition probability is summed to evaluate the model (Chen et al., 2013):
$$Score = \frac{1}{12n}\sum\nolimits_{t-1}^{12}\sum\nolimits_{s=1}^{n} p_{s,t} \qquad (29)$$

where $p_{s,t}$ characterizes the transition probability in month $t$ of year $s$, and $n$

is the length of the validation period. The calculation results are shown in Table 5.

With the same time scale of SPI, the model scores of Model 1 and Model 2

lowers as the forecast period $M$ lengthens, while the model scores of Model 3 are not
significantly affected by the forecast period $M$. Model 1 had the highest rating of 0.36
at an SPI of 1-month time scale and a forecast period of one month; Model 2 reached
the highest model rating of 0.74 at a 12-month time scale and a forecast period of one
month; and model-3 performed well at an SPI of 1-month time scale and a 12-month
time scale. Overall, model-3 has the highest rating, and Model 1 has the lowest rating
for the same SPI time scale and the same forecast period, which also indicates that the
forecast accuracy of the conditional distribution model considering the *HI* is higher
for short-term forecasts with a forecast period of 3 months or less, and involving the
*HI* can further improve the forecast accuracy of the model.

Table 5. Model Evaluation (Model 1: Multivariate normal distribution model; Model 2:

Conditional distribution model; Model 3: Conditional distribution model with *HI*)

| Model type | Lead time *M* | SPI time scale | | | |
|---|---|---|---|---|---|
| | | 1 | 3 | 6 | 12 |
| Model 1 | 1 | 0.36 | 0.36 | 0.28 | 0.22 |
| | 2 | 0.11 | 0.35 | 0.27 | 0.22 |
| | 3 | 0.02 | 0.34 | 0.26 | 0.22 |
| Model 2 | 1 | 0.69 | 0.52 | / | 0.74 |
| | 2 | 0.69 | 0.47 | / | 0.67 |
| | 3 | 0.69 | 0.44 | 0.39 | 0.60 |



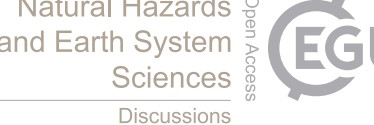

| | | | | | |
|---|---|---|---|---|---|
| | 1 | 0.72 | 0.64 | 0.59 | 0.71 |
| Model 3 | 2 | 0.71 | 0.64 | 0.59 | 0.71 |
| | 3 | 0.72 | 0.64 | 0.60 | 0.71 |

## 5 Conclusions

Many studies have pointed out that human activities have a significant impact on
watershed runoff in the Luanhe River Basin. In this paper, three probability models
were constructed to calculate the transition probabilities from current SPI classes or
values to future SRI classes; then, a scoring mechanism was applied to evaluate the
performance of the models.
Under the condition of considering the *HI*, the calculation results of the drought
transition probability show that when the value of *HI* is less than 0, human activity
slows the occurrence of hydrological drought and the probability of maintaining
hydrological drought at the normal class peaks. With the increase in the *HI* value, it is
easier for hydrological droughts to transition to more severe droughts. The calculation
results of Model 1 and Model 2 show that the future hydrological drought classes are
likely to be the same as the meteorological drought classes in the current period.
Finally, a scoring mechanism was applied to the evaluation of the models, and
the forecast results of the three models were evaluated. The results demonstrate that
when the SPI time scale is the same, the scores of Model 1 and Model 2 lower as the
forecast period lengthens. In most cases, Model 2 performs better than Model 1, and
the performance of Model 3 is the most stable of the three models and has the highest
score. In addition, the performance of Model 3 is not affected by the forecasting
period. The conditional probability model considering *HI* is more suitable for the
Luanhe River basin, where human activities have a high influence.
Although this study has made some progress in the forecasting of hydrological
drought in the changing environment, only one exogenous variable was calculated to
quantify the impact of human activities, and the climate factors can be further
considered in future studies. In addition, *HI* can be analysed specifically, such as land
use and social economy.



---

**Limitation:** Under changing environmental conditions, the driving factors of drought
can be analysed from the physical mechanism, such as considering the influence of
large-scale climate indices or hydro-meteorological variables, to further improve the
forecasting accuracy of hydrological drought.
**Ethical Approval:** This work meets the ethical and moral requirements.
**Consent to Participate:** M L. MF Z. RX C. YD S and XY D all agreed to participate
in the research for the article.
**Consent to Publish:** M L. MF Z. RX C. YD S and XY D all agreed to publish this
article.
**Authors Contributions:**
M L(First Author and Corresponding Author):Conceptualization, Methodology,
Software, Investigation, Formal Analysis, Writing-Original Draft;
MF Z:Data Curation, Writing-Original Draft;
RX C: Visualization, Investigation;
YD S: Resources, Supervision;
XY D: Visualization, Writing-Review & Editing.
**Competing interest:** M L. MF Z. RX C. YD S and XY D all declare that there is no
conflict of interest.
**Data availability statement:** We are grateful to the Hydrology and Water Resource
Survey Bureau of Hebei Province for providing runoff data. The data and materials of
the research are available.

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

**Acknowledgements: This work was supported by the State Key Laboratory**
**of Hydraulic Engineering Simulation and Safety Program (No. HESS-2206**
**and No. HESS-2222).**