# Peer review of "Hydrological drought forecasting under a changing environment"

_Natural Hazards and Earth System Sciences, 2022_

## Referee Comment (RC4)

The manuscript evaluates drought in the Luan River basin by applying well-known indices and proposing a new methodology based on the combination of human activities. The manuscript structure could be more satisfying. The language is poor and needs to be checked. The figure and table are well-readable. Before publication, there are some significant points to be clarified.

1. There is a lack of scientific discussion. The authors briefly introduce some literature findings but need to clearly identify the position of their proposed methodology in the scientific literature. Moreover, why did they choose just SPI and SRI? I expected to see at least the SPEI, which is quite simple as the SPI. Does the catchment have some groundwater influence? If yes, the time window should be extended to 24 or 48 months to consider such an aspect.

2. I don't agree with this definition of the Human Index. It seems speculation to identify a single point transition from "natural" land cover to "artificial" or "human" land cover. Moreover, the hypothesis that 1979 is the changing point should be carefully explained from a physical point of view.

3. This "Human Index" has been built with a simple linear regression. The authors should at least discuss this hypothesis since the rainfall-runoff generation is entirely non-linear.

4. HI < 0 means that the actual SRI is greater than the theoretical SRI without human activities. L176: "HI<0 has the opposite effect" rephrase this sentence. Moreover, looking at Fig .5, the HI's fitted line always starts from a negative value and continues for several years, making the choice of 1979 less reasonable. Finally, why in the mid 90 years, there are always severe negative values? The authors should try to explain this strange behaviour.

5. I suggest significantly improving the explanation of the 4.4 paragraph. I supposed they employed just the verification/validation period to calculate the score. But I'm not sure this is a good evaluation metric since it involves a sum of transition probability. Furthermore, since they proposed a condition probability with a less variable exogen factor, the probability varies less. The authors should better write this paragraph and discuss it.

6. What type of interpolator did they use to obtain the rainfall time series? How is the orography of the catchment? This last information should be inserted in Fig 1.

---

## Author Response (AR1)

*Thanks to the experts for their valuable comments on the article. I have replied to the questions and comments raised by the three experts one by one.*

*Reviewer1:*
*Recommendation: Minor Revisions*
*Comments:*
*I appreciate NHESS for providing the opportunity of reviewing the manuscript. I work in the similar research field and I'm sure that such research has useful meanings in practice and the authors must spare much time finishing the work. This study proposed to use meteorological drought and human activity to predict the hydrological drought. This is a useful study that addresses an important topic. The results of the proposed methodologies were tested and compared with the traditional ones. Overall, the results are reasonable and could support the conclusions. Some issues are listed below.*

*1) The drought prediction methodology is not introduced. The author only briefly touched based on the prediction. Some discussion on the prediction method can beef up this study.*
*Response: Thanks for the expert opinion, we have supplemented the drought prediction methodology and some discussion on the prediction method in the introduction.*

*2) The authors calculated the transition probability of drought. Why the SPI and SRI were chosen instead of other drought indices such as SSI? I recommend the authors to include some explanation of the drought indexes (SPI and SRI) in the introduction.*
*Response: Thanks for the expert opinion, similar to some other hydrological drought indexes such as SSI, SRI can capture the characteristics of hydrological drought, and its calculation is relatively simple. In this paper, we used SPI to capture the characteristics of meteorological drought and used SRI to capture the characteristics of hydrological drought. We have included some explanation of the drought indexes (SPI and SRI) in the introduction.*

*3) Section 4.2. "According to the normality test results of rainfall and runoff series". Since you fit the distribution based on SPI and SRI. They are, by definition, normal distribution and thus should obey the normal distribution. Why bother performing the normality test?*
*Response: Thanks for the expert opinion, here is our clerical error and this sentence has been changed to: "According to the definition, the SPI and SRI series obey the normal distribution." In this study, we did not perform the normality test.*

*4) The first two paragraphs of section 4.4 should be moved to the method section.*
*Response: Thanks for the expert opinion, the first two paragraphs of section 4.4 have been moved to the method section.*

*Reviewer2:*

The manuscript investigates hydrological drought forecasting in the Luanhe River Basin. A conditional distributional model incorporating an established human activity index as the exogenous variable was proposed to predict the transitional probability from meteorological drought to hydrological drought. The results of the proposed methodologies were tested and compared with the traditional ones. Overall, the results are reasonable and could support the conclusions. Some issues are listed below.

1) Throughout the results analysis section, the results are mostly focused on very detailed preliminary results, whereas insights and discussion of results are lacking. It is quite obvious that the transitional probability depends on SPI and the lead time, while the readers might be more interested in the new insights brought by this new methodology. The authors might need to rephrase some of the texts in the results analysis section to make them more logically connected.

**Response**: Thanks for the expert opinion. According to the opinion of experts, we have further summarized the conclusions and added our insights and discussions based on the new methodology.

2) Lines 4-10, page 12, the justification of 1979 as the change point (due to human activity) is not sound enough. Why the sudden change in annual runoff in 1979 is not caused by the heavy rainstorms?

**Response**: Thanks for the expert opinion, according to historical records, local human activities (such as land use change, reservoir construction, etc.) are regarded as the main factor influencing runoff (Yan et al.,2018; Chen et al., 2021).  In this paper, it is determined that 1979 is the change point of the runoff sequence in the basin and this conclusion is consistent with Li et al. (2015) and Wang et al. (2015). We have further described the reasons for the sudden change of runoff series in the manuscript.

3)the authors determine the change point with the Nonstationarity analysis. Do you perform the prediction based on data of each period? In this case, the sample size may be short. How to tackle this problem?

**Response**: Thanks for the expert opinion, in this paper, the study period is divided into a correction period (1961-2003) and a verification period (2004-2010) and we perform the prediction based on data of the correction period (1961-2003).

4)The authors introduce the multivariate distribution model and the conditional model for the prediction. The motivation of this method should be highlighted. For example, there are multiple prediction models out there. Why do the authors select this model? This model is closely associated with the copula mode. What is the difference or why do you select this model instead of other models?

**Response**: Thanks for the expert opinion, the Copula can be adopted to model the dependence structure between meteorological/hydrological drought indices, and based on the conditional probability, the transition probabilities and transition thresholds from different classes of meteorological drought to hydrological drought were calculated. (Majid et al.,2019). The traditional probability prediction models (such as the Multivariate normal distribution model, Markov Model, etc) can be used to

*calculate the transition probabilities from the current drought index classes to the future drought classes, but the conditional probability model can calculate the transition probabilities from the current drought index values to the future drought classes, which is more robust to forecast hydrological drought than the traditional probability prediction models. We have highlighted the motivation of the conditional distribution model in the introduction according to expert opinion.*

*5)Lines 9, page 15, "Transition probabilities involving HI", did you mean "Transition probabilities with involving HI as the covariate"?*
***Response**: Thanks for the expert opinion, in order to express our meaning more clearly, the sentence in the text has been changed to: "Transition probabilities with involving HI as the covariate".*

*6)There are some language issues in this manuscript, a thorough editorial check might be needed.*
***Response**: Thanks for the expert opinion, we have carefully examined and revised the language problems in the manuscript.*

*Reviewer3:*

*The manuscript evaluates drought in the Luan River basin by applying well-known indices and proposing a new methodology based on the combination of human activities. The manuscript structure could be more satisfying. The language is poor and needs to be checked. The figure and table are well-readable. Before publication, there are some significant points to be clarified.*

*1) There is a lack of scientific discussion. The authors briefly introduce some literature findings but need to clearly identify the position of their proposed methodology in the scientific literature. Moreover, why did they choose just SPI and SRI? I expected to see at least the SPEI, which is quite simple as the SPI. Does the catchment have some groundwater influence? If yes, the time window should be extended to 24 or 48 months to consider such an aspect.*
***Response**: Thanks for the expert opinion, we have adjusted some position of the drought prediction methodology and some discussion on the prediction method in the introduction. Moreover, both SPEI and SPI can capture the characteristics of meteorological drought. As an ideal evaluation index of meteorological drought, SPEI is calculated by using the difference between precipitation and potential evapotranspiration (PET) to characterize regional drought. SPI is only calculated based on precipitation series and its calculation is relatively simple. At present, we used SPI to capture the characteristics of meteorological drought in this paper, and the effects of SPEI and SPI on drought propagation can be further compared and analyzed in future studies.*

*Influenced by topography, meteorology, hydrology and hydrogeological conditions, the spatial distribution of groundwater resources in the Luanhe River basin*

*is quite different. The recharge and storage conditions of shallow groundwater in plain areas and intermountain basins are relatively superior, and the content of groundwater in mountainous areas is relatively small (the area of mountainous areas in the Luanhe River basin accounts for 98.2%). Therefore, the total amount of water resources in the Luanhe River basin is mainly considered to be affected by the amount of surface water resources. We mainly consider the surface runoff in this paper, the runoff data from 1961 to 2010 come from the measured stations and the SRI can be calculated for 1-month, 3-month, 6-month, and 12-month time scales to characterize hydrological drought based on these data.*

*2) I don't agree with this definition of the Human Index. It seems speculation to identify a single point transition from "natural" land cover to "artificial" or "human" land cover. Moreover, the hypothesis that 1979 is the changing point should be carefully explained from a physical point of view.*

**Response**: *Thanks for the expert opinion, in this paper, the double cumulative curve of precipitation-runoff was used to identify the change point of watershed runoff series. The effects of human activities are complex. In order to quantify the impact of human activities, the change point was identified, and then it was approximately believed that the difference in the relationship between precipitation and runoff before and after the change point was caused by human activities. Moreover, the HI is easy to calculate and can approximately replace the influence of human activities. According to historical records, local human activities (such as land use change, reservoir construction, etc.) are regarded as the main factor influencing runoff (Yan et al.,2018; Chen et al., 2021). We have further explained the reasons for the change of runoff series from a physical point of view in Section 4.1 of the manuscript.*

*3) This "Human Index" has been built with a simple linear regression. The authors should at least discuss this hypothesis since the rainfall-runoff generation is entirely non-linear.*

**Response**: *Thanks for the expert opinion. The double cumulative curve method can test the nonstationarity of the bivariate correlation between rainfall series and runoff series, and the point where the underlying surface was significantly altered by human activities can be determined according to the position of the slope change of the curve. Due to the short data series before and after the change point (20 years before the change point and 30 years after the change point), the linear equation was approximately used to fit the relationship of precipitation and runoff, which we have explained in Section 3.2 of the manuscript.*

*4) HI < 0 means that the actual SRI is greater than the theoretical SRI without human activities. L176: "HI<0 has the opposite effect" rephrase this sentence. Moreover, looking at Fig .5, the HI's fitted line always starts from a negative value and continues for several years, making the choice of 1979 less reasonable. Finally, why in the mid 90 years, there are always severe negative values? The authors should try to explain this strange behaviour.*

***Response****: Thanks for the expert opinion, the sentence: "HI<0 has the opposite effect" has been rephrased to "HI < 0 means that the actual SRI is greater than the theoretical SRI without human activities".*

*Moreover, as can be seen from Figure 5, the abscissa in the figure is from 1979 year, the values of HI were considered to be 0 before 1979 year. In fact, although the fitted line starts with some negative values, HI values does not always start from a negative value. The fitting line does not represent the HI value, but mainly indicates that HI has a significant upward trend.*

*Finally, according to historical statistics, a large number of water projects were built and operated between 1990 and 2000.The storage capacity of water conservancy projects increased from 259 million $m^2$ to 351 million $m^3$. Therefore, the construction and operation of large reservoirs in the mid-1990s may be the main reason for the serious negative values of HI. We have explained in the manuscript.*

*5) I suggest significantly improving the explanation of the 4.4 paragraph. I supposed they employed just the verification/validation period to calculate the score. But I'm not sure this is a good evaluation metric since it involves a sum of transition probability. Furthermore, since they proposed a condition probability with a less variable exogen factor, the probability varies less. The authors should better write this paragraph and discuss it.*

***Response****: Thanks for the expert opinion, in order to evaluate and compare the performance of the three models, we divided the whole study period into two parts: calibration period (1960–2003) and validation period (2004–2011). Based on the measured data, it is shown that the larger the sum of transition probability is, the more accurate the prediction result is.*

*Furthermore, the calculation results of the model involving HI as an exogenous variable are significant different to the models without considering human activities, and we have supplemented the discussion in the text:*

*The calculation results of Model 1 and Model 2 show that, the future hydrological drought classes are more likely to be the same as the meteorological drought classes in the current period, and it is more significant in the Model 2. In addition, it is obvious that drought transition probabilities of the Model 3 are significant different to the Model 1 and Model 2. Take the figure 6 (b) as an example,when $z_0 = -1.25$ and $C_0 = Mo$, the result of Model 1 shows that the probability of SPI values transit to SRI classes in the future month in the normal class is close to 0.15, the result of Model 2 is close to 0, but the result of Model 3(HI=0) is close to 0.95. The results of Model 3(HI=0) indicate that the hydrological drought is likely to remain at the normal class in the future month. Moreover, the value of HI also has a great impact on the results of model 3, for example, when HI=−2 or −1, the probabilities of SPI values transit to SRI classes in the future*

*month in the normal class are both close to 1, but the probability is close to 0.65 and 0.17 respectively when HI=1 and 2.*

*6) What type of interpolator did they use to obtain the rainfall time series? How is the orography of the catchment? This last information should be inserted in Fig 1.*

**Response:** *Thanks for the expert opinion, the average monthly rainfall data of the area are obtained by Inverse-Distance-Weighting interpolation method, and we have explained in the manuscript. The topographic characteristics of the catchment area have been supplemented in the Section 2 of the text, and the geographic characteristics of the basin have been inserted in Fig 1.*

*References*

*Yan Xiaolin,Bao Zhenxin,Zhang Jianyun,Wang Guoqing,He Ruimin,Liu Cuishan. Quantifying contributions of climate change and local human activities to runoff decline in the upper reaches of the Luanhe River basin[J]. Journal of Hydro-environment Research, 2018, 28(C) : 67-74.*

*Chen, Xu,Han, Ruiguang,Feng, Ping,Wang, Yongjie. Combined effects of predicted climate and land use changes on future hydrological droughts in the Luanhe River basin, China[J]. Natural Hazards,  2021, : 1-33.*

---

## Author Response (AR2)

*I appreciate the efforts conducted by the authors since they tried to answer all my points. While I'm not completely convinced by it, I recommend it for publication after a few minors.*

*1.Table 4: missing. Probably table 5 should be renamed to table 4*

**Response**: Thanks for the expert opinion, the table 5 has been renamed to table 4.

*2.Figure1 The names of the rain gauge stations are not visible; I suggest the authors use different font colours; moreover, in the legend DEM in m asl.*

**Response**: Thanks for the expert opinion, in order to make the name of the rainfall station clear, Figure 1 has been readjusted. Moreover, in the legend DEM in m asl.

[Figure]

Figure 1 The geographical location of the Luanhe River basin

*3.Table 3: For better readability, I suggest renaming the month name in Jan, Feb, Mar, etc.., instead of just using the first letter.*

**Response**: Thanks for the expert opinion, the month name in Table 3 has been changed to Jan, Feb, Mar, Apr, May, Jun, Jul, Aug, Sept, Oct, Nov and Dec.

Table 3. *HI*-12 monthly mean and standard deviation

|  | Jan | Feb | Mar | Apr | May | Jun | Jul | Aug | Sept | Oct | Nov | Dec |
|---|---|---|---|---|---|---|---|---|---|---|---|---|
| Mean | -0.04 | -0.03 | -0.03 | -0.03 | -0.03 | 0.00 | 0.06 | 0.06 | 0.10 | 0.10 | 0.09 | 0.06 |

| Sd | 1.36 | 1.37 | 1.38 | 1.41 | 1.41 | 1.51 | 1.40 | 1.40 | 1.45 | 1.44 | 1.44 | 1.43 |

*4.Figure captions: erase brackets in numbers.*

**Response**: Thanks for the expert opinion, the brackets of the Figure captions have been erased.